# Click Decoration of *Bombyx mori* Silk Fibroin for Cell Adhesion Control

**DOI:** 10.3390/molecules25184106

**Published:** 2020-09-08

**Authors:** Hidetoshi Teramoto, Minori Shirakawa, Yasushi Tamada

**Affiliations:** 1Silk Materials Research Unit, Division of Biotechnology, National Agriculture and Food Research Organization (NARO), 1-2 Owashi, Tsukuba, Ibaraki 305-8634, Japan; 2Faculty of Textile Science and Technology, Shinshu University, 3-15-1 Tokida, Ueda, Nagano 386-8567, Japan; 14f4031h@shinshu-u.ac.jp

**Keywords:** *Bombyx mori*, cell patterning, click chemistry, genetic code expansion, silk fibroin, transgenic silkworm

## Abstract

Silk fibroin produced by the domesticated silkworm, *Bombyx mori*, has been studied widely as a substrate for tissue engineering applications because of its mechanical robustness and biocompatibility. However, it is often difficult to precisely tune silk fibroin’s biological properties due to the lack of easy, reliable, and versatile methodologies for decorating it with functional molecules such as those of drugs, polymers, peptides, and enzymes necessary for specific applications. In this study we applied an azido-functionalized silk fibroin, *AzidoSilk*, produced by a state-of-the-art biotechnology, genetic code expansion, to produce silk fibroin decorated with cell-repellent polyethylene glycol (PEG) chains for controlling the cell adhesion property of silk fibroin film. Azido groups can act as selective handles for chemical reactions such as a strain-promoted azido-alkyne cycloaddition (SPAAC), known as a click chemistry reaction. We found that azido groups in *AzidoSilk* film were selectively decorated with PEG chains using SPAAC. The PEG-decorated film demonstrated decreased cell adhesion depending on the lengths of the PEG chains. Azido groups in *AzidoSilk* can be decomposed by UV irradiation. By partially decomposing azido groups in *AzidoSilk* film in a spatially controlled manner using photomasks, cells could be spatially arranged on the film. These results indicated that SPAAC could be an easy, reliable, and versatile methodology to produce silk fibroin substrates having adequate biological properties.

## 1. Introduction

Silk fibroin is a huge protein polymer produced by the domesticated silkworm *Bombyx mori*. It is regarded as a heterodimer of fibroin heavy chain (FibH; ~390 kDa) and fibroin light chain (FibL; ~26 kDa) [1,2]. The amino acid sequence of FibH is dominated by Gly- and Ala-rich repetitive motifs such as Gly-Ala-Gly-Ala-Gly-Ser [1]. Due to its strong self-assembling property, silk fibroin can be processed into self-standing materials in various forms such as film, hydrogel, nonwoven nanofiber mats, and porous sponge from aqueous solution without chemical crosslinking [3]. Since these silk fibroin materials combine mechanical strength, biocompatibility, and biodegradability, they work as good substrates for various types of cells [4,5,6]. On the other hand, it is often difficult to precisely optimize the biological properties of silk fibroin for specific biomedical applications because there are no easy, reliable, and versatile methodologies to decorate it with desired foreign components such as drugs, polymers, peptides, and enzymes.

Decoration of silk fibroin has already been realized by conventional chemical modification techniques [7,8], although such chemical procedures often raise concerns about contamination by toxic reagents as well as the degradation of silk fibroin molecules. Recently we have proposed a novel strategy to decorate silk fibroin based on an emerging biotechnology called genetic code expansion.

The standard genetic code, which is virtually universal in all life forms on Earth, specifies 20 amino acids that can be used in protein synthesis. The genetic code expansion methodology can expand the repertoire of amino acids for protein synthesis, thus enabling the creation of proteins with novel structures and functions [9,10,11,12]. Starting from unicellular organisms such as bacteria and yeast, the genetic code expansion has been applied to multicellular animals such as flies and mice [13,14].

A research group at NARO first applied the genetic code expansion methodology to *B. mori* in order to create silk fiber with novel functions [15]. They succeeded in efficiently incorporating an azido-bearing synthetic amino acid, 4-azido-l-phenylalanine (AzPhe), into silk fibroin in collaboration with a research group at RIKEN [16]. Azido is not found in natural biomolecules and can be modified in a highly selective manner by click chemistry reactions under moderate conditions [17]. Silk fibroin with azido functionalities, which is designated as *AzidoSilk*, can thus be decorated in an easier and more reliable manner compared to the conventional chemical modification methods [18]. Protein modification by click chemistry reactions is known to be compatible with live cells [19], indicating the low toxicity of click chemistry reactions to living systems. These benefits of click chemistry reactions strongly indicated that silk fibroin decorated using this novel approach would be especially useful for the development of silk-based biomaterials. In addition, an azido group directly connected to an aromatic group is photolabile [20]. Using this property, the spatial arrangement of functional molecules on *AzidoSilk* matrices has been demonstrated via the partial decomposition of azido groups by UV irradiation followed by click chemistry reactions [18].

Recently, *AzidoSilk* production was scaled up by a simple hybridization of a transgenic line that has the ability to incorporate AzPhe in silk fibroin with a high-silk-producing strain [21]. This achievement, combined with various functionalization approaches by click chemistry reactions, would facilitate industrial applications of silk fibroin materials with specific functions.

In this study, we attempted to alter the cell adhesion property of *AzidoSilk* film by click chemistry reactions with polyethylene glycol (PEG), which is known to repel proteins and cells. The PEG-modified *AzidoSilk* film showed decreased adhesion of mouse fibroblasts. By partial UV irradiation through photomasks before click reactions with PEG, the spatial arrangement of cells on *AzidoSilk* film was achieved. These achievements by the power of genetic code expansion would lead to the production of self-standing, biocompatible, and spatially arrangeable silk-based cell substrates for tissue engineering applications.

## 2. Results and Discussion

### 2.1. Click Decoration of Silk Fibroin with PEG

We prepared *AzidoSilk* by feeding an artificial diet containing AzPhe (0.05% in dry diet) to fifth-instar larvae of the H06 transgenic *B. mori* line, which expresses the F432V mutant of *B. mori* phenylalanyl-tRNA synthetase α-subunit (BmPheRS-α) in its posterior silk glands [16]. Wildtype *B. mori* larvae are not able to use AzPhe as a building block in protein synthesis because AzPhe cannot be a substrate of wildtype BmPheRS-α. On the other hand, the F432V mutant of BmPheRS-α can recognize AzPhe as a substrate to aminoacylate tRNA^Phe^ with it, which enables the translation of some Phe codons as AzPhe in a random manner in the posterior silk glands of the H06 transgenic line [16]. The *AzidoSilk* used in this study was estimated to contain approximately two AzPhe residues per silk fibroin molecule, which is comprised of ca. 5500 amino acids [16]. The number of AzPhe residues in *AzidoSilk* was estimated by high-resolution mass analyses of a Phe-containing peptide fragment digested from FibL. Azido groups on these AzPhe residues can act as selective handles for chemical modifications [16,18]. During the alkaline degumming process of the cocoons with Na_2_CO_3_ to remove sericin coating, fibroin molecules were hydrolyzed to smaller molecular weights with a broad distribution centered at around 150–250 kDa as shown in Appendix A. Azido groups in *AzidoSilk* retained their SPAAC (strain-promoted azido-alkyne cycloaddition) reactivity after the degumming process (Appendix A), thus demonstrating that the functionalization of *AzidoSilk* by SPAAC is compatible with the standard purification process of silk fibroin for biomaterials development [3].

*AzidoSilk* was first processed into aqueous solution and coated on cell culture plates as a transparent film by conventional methods [3,18]. The *AzidoSilk* film was then reacted with dibenzylcyclooctyne (DBCO)-functionalized methylated PEG reagents (DBCO-mPEG) of 5, 10, 20, and 30 kDa chain lengths. A DBCO group selectively reacts with an azido group without catalysts (Figure 1). This type of reaction between an azido and a cyclic alkyne is called strain-promoted azido-alkyne cycloaddition (SPAAC). Since SPAAC requires no metal catalyst, it is appropriate for biomedical applications. To find a condition where almost all azido groups are consumed during the reaction, the concentrations of DBCO-mPEGs were changed from 30 to 300 μM (Figure 2). The unreacted azido groups after the reactions were reacted with a DBCO-functionalized biotin reagent by SPAAC, and the attached biotins were detected with horseradish peroxidase-conjugated streptavidin (HRP-streptavidin) (Figure 2). The signals due to biotin attachments to the unreacted azido groups gradually decreased along with the increase in the concentration of DBCO-mPEG. The signals from the control samples were probably due to nonspecific binding of the biotin reagent. The results demonstrated that 300 μM DBCO-mPEG was enough for reactions with almost all azido groups in *AzidoSilk* film regardless of its chain length.

### 2.2. Cell Adhesion on PEG-Decorated Silk Fibroin Film

Initial adhesion of mouse fibroblasts (NIH3T3) after 2 h culture on the surface of normal silk fibroin and *AzidoSilk* was investigated when treated with DBCO-mPEG of different chain lengths. We used a serum-free medium for the test to emphasize the behavior of cells in their initial adhesion period by minimizing the influence of serum on cell adhesion. Figure 3 shows representative pictures of cells attached on each surface. Compared with cells cultured in a serum-containing medium (Appendix A), cells with spherical forms were abundant. Some of the cells showed spread forms on nontreated surfaces of both normal silk fibroin and *AzidoSilk*. When cells were treated with DBCO-mPEG, their morphology was different between normal silk fibroin and *AzidoSilk*. Cells on *AzidoSilk* film mostly showed circular forms after PEG treatments, especially when longer PEG chains were used. On the other hand, cells with spread forms were found on normal silk fibroin even after the treatment with DBCO-mPEG of 30 kDa chain length. These differences in cell morphology would be due to the decoration of *AzidoSilk* with PEG chains, which makes the surface of silk fibroin more repellent to proteins necessary for cell attachment than the nondecorated surface.

Figure 4 shows the average number of adherent cells. Nontreated surfaces showed good attachment of fibroblasts, comparable to that of tissue-culture-treated polystyrene (TCPS). The adhesion of fibroblasts on normal silk fibroin showed a tendency to decrease depending on the chain length of DBCO-mPEG: treatment with longer chains of DBCO-mPEG led to greater decreases in cell number. The decrease in cell number was more remarkable for *AzidoSilk*. Statistical analysis (Tukey–Kramer test) showed a significant difference between them when treated with DBCO-mPEG of 20 and 30 kDa chain lengths.

Normal silk showed decreased cell adhesion when treated with DBCO-mPEG, which would be due to nonspecific binding of the reagents. We assumed that hydrophobic DBCO groups in DBCO-mPEG nonspecifically bound to silk fibroin through the noncovalent binding, although no significant change was observed in the water contact angle of normal silk fibroin film due to the presence of PEG chains on its surfaces (Appendix A). The cell adhesion on *AzidoSilk* was significantly lower than that on normal silk when treated with DBCO-mPEG of 20 and 30 kDa chain lengths. This significant difference arose from covalent binding of the reagents on *AzidoSilk* by selective reaction between azido groups and DBCO. To confirm the presence of PEG chains on the film surface, water contact angles were measured after treatment with DBCO-mPEG of 20 kDa chain length (Appendix A). When *AzidoSilk* film was treated with DBCO-mPEG, a significant decrease in the contact angle was observed, confirming the attachment of PEG chains on the surface of *AzidoSilk* film. As shown in Figure 2, the click reaction was almost completed for DBCO-mPEG of all chain lengths, indicating that the molar ratio between the silk molecule and the PEG chain was approximately equal for all chain lengths. It is considered that longer chains could cover larger surface areas than shorter ones when the molar amounts of decoration are the same. Since longer PEG chains exhibited more significant effects on cell repulsion, we used DBCO-mPEG of 20 kDa chain length for subsequent experiments.

### 2.3. Spatial Patterning of Cells on Partially PEG-Decorated Silk Fibroin Film

Aromatic azido groups such as that in AzPhe are known to decompose upon UV irradiation. Since azido groups in *AzidoSilk* film in UV-irradiated areas decompose and no longer act as chemical handles for click chemistry reactions, patterned decoration of functional molecules can be achieved by click reaction after partial UV irradiation through photomasks [18]. It was thus expected that fibroblasts could also be patterned on *AzidoSilk* film by partial decoration of PEG chains.

*AzidoSilk* films formed on the bottoms of 60-mm-diameter culture dishes were first irradiated by 254 nm UV light for 1.5 min through patterned photomasks (line and alphabetical patterns, Figure 5B,E) followed by modification with DBCO-mPEG of 20 kDa chain length. The irradiation time was set to 1.5 min based on an independent UV irradiation test using fibroin aqueous solution (Appendix A). The result demonstrated that 1 min irradiation time was enough to decrease the signals from azido groups to the background level. The photomask was a negative type where azido groups in the exposed areas were expected to decompose and lose their reactivity with DBCO-mPEG (Figure 5A). Mouse fibroblasts were cultured in the dishes for 3 days. The boundary between UV-irradiated and nonirradiated areas was observed on the third day of culture (Figure 5C,F). Cells were confluently grown in the UV-irradiated area where click reactions with DBCO-mPEG did not occur. On the other hand, much smaller numbers of cells were observed in the nonirradiated areas where silk fibroin was decorated with PEG chains. Wide views of the dishes after cell fixation with methanol showed that the patterns of the photomasks were successfully transferred to the patterns made by cells on silk fibroin film (Figure 5D,G). Azido groups under the openings in the photomasks (Figure 5B,E) decomposed and lost their reactivity with DBCO-mPEG. *AzidoSilk* with decomposed azido groups thus showed the intrinsic cell adhesion property of silk fibroin. In contrast, uniform proliferation of the cells was observed without PEG decoration (Appendix A), and extremely small numbers of cells were observed on the PEG-decorated surface without UV irradiation (Appendix A). These results demonstrated that click decoration of silk fibroin with PEG chains can actually control cell adhesion on silk fibroin substrate in a biocompatible manner. The confluent cell growth on *AzidoSilk* without PEG decoration (Appendix A, left panels) indicated that the existence of AzPhe does not cause serious adverse effects on the biocompatibility of silk fibroin.

Several previous works reported the achievement of cell patterning using silk fibroin as a substrate based on conventional photolithographic methods [22,23,24]. These works achieved excellently precise patterning of silk fibroin itself. Most of the relevant studies use inorganic substrates such as silicon and glass to support patterned silk fibroin. A unique approach was reported by Xu et al. [24], who used crosslinked silk fibroin film as a substrate instead of inorganic substrates to provide self-standing patterned silk fibroin materials. The method used in the present study is not able to pattern silk fibroin itself but can pattern functional molecules directly on fibroin film by controlling the spatial presence of azido groups as chemical handles for click chemistry reactions with cell-repellent PEG reagents. One advantage of our approach is that we could prepare self-standing film substrates with cell patterning ability using only silk fibroin without any other substrates for biomedical applications.

## 3. Materials and Methods

### 3.1. Materials and Animals

All chemicals used in this study were of reagent grade and used as received. AzPhe was from Watanabe Chemical Industries (Hiroshima, Japan). DBCO-mPEG (5, 10, 20, and 30 kDa) was from Click Chemistry Tools (Scottsdale, AZ, USA). Sulfo-DBCO-biotin was from Sigma-Aldrich (St. Louis, MO, USA). Cellulose powder, HRP-streptavidin, and dimethyl sulfoxide (DMSO) were from Nacalai Tesque (Kyoto, Japan). TMB 1-Component Microwell Peroxidase Substrate (SureBlue) was from SeraCare Life Sciences (Milford, MA, USA). Mouse fibroblast NIH3T3 cells, provided by the RIKEN BRC, were used for cell adhesion and proliferation assays. The H06 transgenic *B. mori* line, which expresses the F432V mutant of BmPheRS-α in its posterior silk glands, was used for the production of *AzidoSilk* [16]. *B. mori* larvae were reared on an artificial diet (SilkMate PS) (Nosan, Yokohama, Japan) at 22–26 °C.

### 3.2. Production of AzidoSilk

AzPhe was mixed with an artificial diet in dried form (SilkMate PM) (Nosan) at ratios of 0.00 (control) and 0.05 wt %. Cellulose powder was added to the control diet to normalize the dry weight. Deionized water of 2.6 times volume per unit weight of dry diet was added and heated for 1 h at 50 °C with gentle shaking to make uniform slurries. The slurries were cooked for 5–10 min at 95 °C in an autoclave and stored in a refrigerator until used. The AzPhe-mixed diet was administered to male larvae of the H06 line from the third day of the fifth instar until spinning began. When the average body weights of the larvae stopped increasing, each larva was individually transferred into a handmade paper box to allow it to start spinning to make a cocoon (at the seventh day in most cases). The harvested cocoons were cut into small pieces and degummed by boiling in 0.02 M Na_2_CO_3_ for 30 min according to a previous paper [3]. This degumming condition is considered a standard purification protocol for the biomedical use of silk fibroin. The degummed cocoons (control or *AzidoSilk*) were dried and stored at −20 °C until used.

### 3.3. Preparation of Fibroin Aqueous Solution

An aqueous solution of fibroin (control or *AzidoSilk*) was prepared according to the previous paper [18]. Briefly, the degummed cocoons were dissolved into 8 M LiBr solution at 35 °C (50 mg/mL) and then mixed with a 1/4 volume of 0.5 M Gly-NaOH buffer (pH 9). The mixture was dialyzed against deionized water and then against 0.1 mM sodium carbonate buffer (pH ~9). The dialyzed silk fibroin solution was centrifuged (9500× *g*, 4 °C, 1 h) to remove insoluble precipitates, and the supernatant was stored at 4 °C until used.

### 3.4. Preparation of Fibroin Film-Coated Cell Culture Plates and Dishes

The fibroin aqueous solution (control or *AzidoSilk*) was diluted to 0.1 wt % with deionized water. For reaction analyses with DBCO-mPEG, 50 μL/well of the diluted solution was put into 96-well cell culture plates (Nunc 167008) and dried at 50 °C. For cell adhesion tests, 150 μL/well of the diluted solution was put into 48-well cell culture plates (Nunc 150687) and dried at 50 °C. For cell patterning tests and contact angle measurements, 3 mL of the diluted solution was put into 60-mm-diameter cell culture dishes (Nunc 150326) and dried at 50 °C. Fibroin films coated on the plates and dishes were insolubilized by water vapor annealing at room temperature for 24 h followed by drying at room temperature [25]. The prepared plates and dishes were stored at room temperature until use.

### 3.5. Click Decoration Tests

To the fibroin (normal silk fibroin or *AzidoSilk*)-coated 96-well plates, 50 μL/well of DBCO-mPEG (0, 30, 75, 150, and 300 μM) of different chain lengths (5, 10, 20, and 30 kDa) dissolved in 50 mM Tris-HCl (pH 8)/50% (*v*/*v*) DMSO was added. The amount of fibroin coated in a single well was 50 μg. When the molecular weight of silk fibroin was estimated to be roughly 400,000, its molar amount in a single well was around 125 pmol. The molar ratios of silk fibroin to DBCO-mPEG of 30, 75, 150, and 300 μM were thus calculated to be, respectively, 1 to 12, 1 to 30, 1 to 60, and 1 to 120. Considering that *AzidoSilk* has approximately two AzPhe residues [16], the concentrations of DBCO-mPEG in the reactions were sufficiently excessive. The plates were sealed with an aluminum seal and incubated overnight at 50 °C. After removing the DBCO-mPEG solution, the wells were washed with 50% (*v*/*v*) DMSO. Then, 50 μL/well of 100 μM sulfo-DBCO-biotin in 50 mM Tris-HCl (pH 8)/50% (*v*/*v*) DMSO was added to each well. The plates were incubated at room temperature for 2 h to label the residual azido groups with biotins. After the sulfo-DBCO-biotin solution was discarded, the wells were washed with DMSO and then with Tris-buffered saline containing 0.05% Tween 20 (TBS-T). To each well, 100 μL/well of HRP-streptavidin (1/5000 dilution in TBS-T) was added followed by incubation at room temperature for 1 h. After the HRP-streptavidin solution was discarded, the wells were thoroughly washed with TBS-T. To each well, 100 μL/well of SureBlue reagent was added followed by incubation at room temperature for 30 min. Coloring was stopped by adding 100 μL/well of 1 M HCl. Absorbance at 450 nm was detected for each well with a VersaMax microplate reader (Molecular Devices, San Jose, CA, USA).

### 3.6. Cell Adhesion Tests

To a 48-well plate coated with fibroin (normal silk fibroin or *AzidoSilk*), we added 150 μL/well of DBCO-mPEG (0 and 300 μM) of different chain lengths (5, 10, 20, and 30 kDa) dissolved in 50 mM Tris-HCl (pH 8)/50% (*v*/*v*) DMSO. The bottom surface area of the 48-well plate was roughly three times that of the 96-well plate. Hence, a three-times-larger volume of reaction solution was used to maintain the molar ratio between silk fibroin and DBCO-mPEG. The plates were sealed with an aluminum seal and incubated overnight at 50 °C. After the DBCO-mPEG solution was removed, the wells were washed with DMSO, 50% (*v*/*v*) DMSO, and deionized water and dried at room temperature. To the fibroin-coated 48-well plate after PEG modification, NIH3T3 cells (5 × 10^4^/cm^2^) were seeded in each well and incubated for 2 h at 37 °C with 5% CO_2_ in serum-free Eagle MEM medium. The morphology of the cells was observed with an IX70 phase contrast microscope (Olympus, Tokyo, Japan). The medium was removed, and the wells were washed with PBS(-) to remove non-adherent cells. The adherent cells in each well were counted by LDH activity assay [26].

### 3.7. Cell Patterning Tests

UV light of 254 nm was irradiated to the *AzidoSilk*-coated 60 mm dish through photomasks with line and alphabetical patterns for 1.5 min using a TL-2000 translinker (UVP, Upland, CA, USA). To the *AzidoSilk*-coated 60 mm dish after UV irradiation was added 1.5 mL/dish of DBCO-mPEG (0 and 300 μM) of 20 kDa chain length dissolved in 50 mM Tris-HCl (pH 8)/50% (*v*/*v*) DMSO. The molar ratio of silk fibroin to DBCO-mPEG of 300 μM was calculated to be 1 to 60, which was still sufficiently excessive. The dishes were incubated overnight at 50 °C. After removal of the DBCO-mPEG solution, the wells were washed with 50% (*v*/*v*) DMSO and deionized water and dried at room temperature. NIH3T3 cells (5 × 10^3^/cm^2^) were seeded on each fibroin-coated 60 mm dish after PEG modification and cultured for 3 days at 37 °C with 5% CO_2_ in Eagle MEM medium containing 10% FBS. Pictures of the cells were recorded with an IX70 phase-contrast microscope. The medium was removed and the dishes were washed with PBS(-). The cells were immobilized by methanol and dried to make them more distinguishable for wide-view observation at low magnification, because cells turn white after methanol fixation. Wide views of the immobilized cells in the dishes were observed with a BZ-X710 inverted microscope (Keyence, Osaka, Japan).

## 4. Conclusions

In this study, we demonstrated the power of genetic code expansion methodology to prepare silk fibroin for decoration by SPAAC with functional molecules in a biocompatible manner. We used *AzidoSilk*, where the azido-bearing synthetic amino acid AzPhe is incorporated in the primary structure of silk fibroin, for the decoration with PEG by SPAAC. Since SPAAC proceeds without catalysts and generates no byproducts, it is ideal for biomedical applications. The PEG-decorated *AzidoSilk* film showed decreased adhesion of NIH3T3 mouse fibroblasts depending on the chain length of PEG (Figure 4). We expect that the same strategy could be extended to other functional molecules such as those of drugs, polymers, peptides, and enzymes because click chemistry reactions are compatible with virtually any type of functional molecule [17,27,28].

An azido group directly attached to an aromatic ring such as in AzPhe is known to decompose by photolysis upon UV irradiation. Using this characteristic, we attempted to spatially control cell adhesion on silk fibroin film. PEG decoration of *AzidoSilk* film after partial UV irradiation through photomasks followed by cell culturing resulted in the formation of patterns made by densely assembled cells (Figure 5). Such methods to spatially arrange cells on substrates are quite useful for the fabrication of various tissue models.

We expect that *AzidoSilk* could be used as a universal platform to produce silk-based biomaterials for a wide range of applications because it is ready for chemical decorations with virtually any type of functional molecule in a biocompatible manner. The successful application of the genetic code expansion methodology to *B. mori* in 2014 [15] made it possible to propose a novel strategy for the alteration of the biological properties of *B. mori* silk fibroin in this study. In this context, further research to expand the repertoire of synthetic amino acids that can be incorporated into silk fibroin would lead to further breakthroughs toward its biomedical applications.

## Figures and Tables

**Figure 1 molecules-25-04106-f001:**
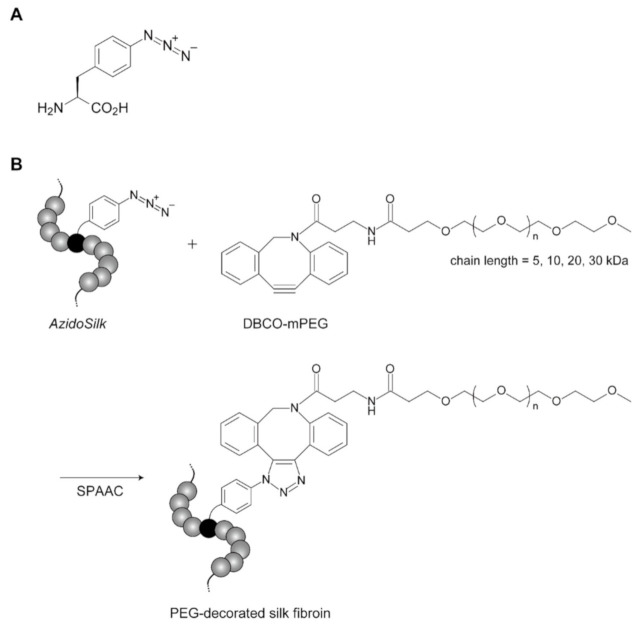
Click decoration of *AzidoSilk* with PEG (polyethylene glycol) chains by SPAAC (strain-promoted azido-alkyne cycloaddition) between an azido group and a DBCO (dibenzylcyclooctyne) group. (**A**) Structure of AzPhe (4-azido-l-phenylalanine). (**B**) Reaction scheme of *AzidoSilk* with DBCO-mPEG (DBCO-functionalized methylated PEG reagents) by SPAAC.

**Figure 2 molecules-25-04106-f002:**
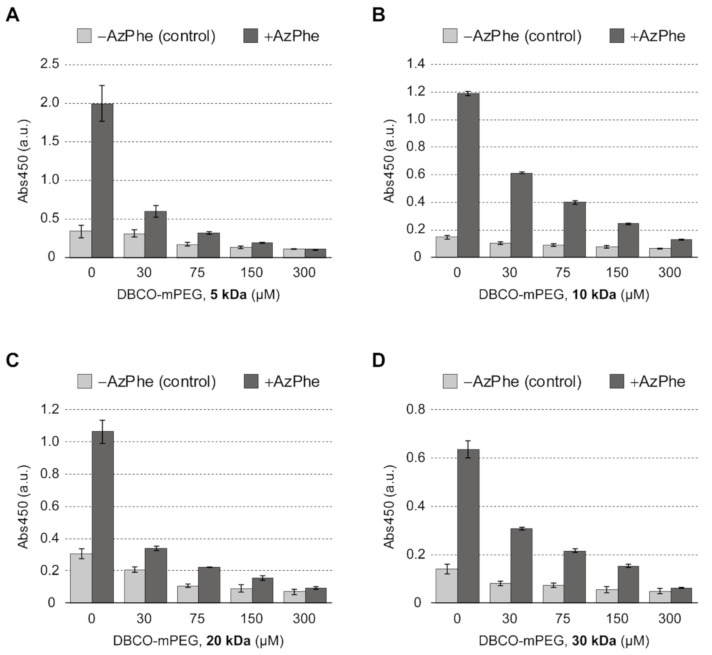
Detection of unreacted azido groups in *AzidoSilk* film on cell culture plates after SPAAC reactions with DBCO-mPEG of different chain lengths: (**A**) 5 kDa, (**B**) 10 kDa, (**C**) 20 kDa, and (**D**) 30 kDa. The concentrations of DBCO-mPEGs were changed from 30 to 300 μm to find an appropriate reaction condition under which to achieve saturation of reactions with DBCO-mPEG. Unreacted azido groups were reacted with sulfo-DBCO-biotin. Biotins were detected by coloration with an HRP substrate after conjugation with HRP-streptavidin. The averaged UV absorbance at 450 nm from four wells was plotted with error bars of standard deviation.

**Figure 3 molecules-25-04106-f003:**
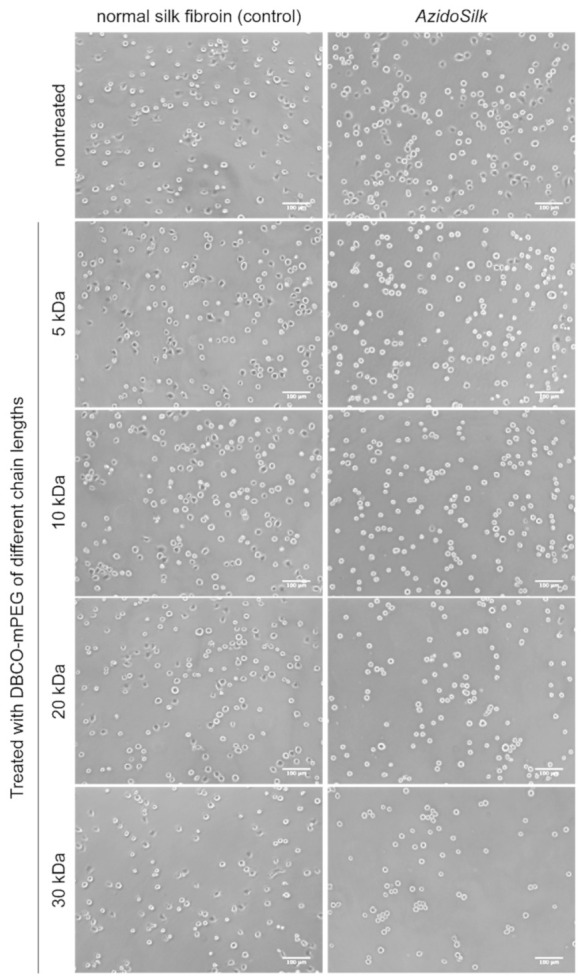
Adhesion of NIH3T3 fibroblasts on the surfaces of normal silk fibroin and *AzidoSilk* film when treated with DBCO-mPEG of different chain lengths (5, 10, 20, and 30 kDa). Pictures were taken after 2 h culture at 37 °C in Eagle minimum essential media (MEM) without fetal bovine serum (FBS). Scale bars are 100 μm.

**Figure 4 molecules-25-04106-f004:**
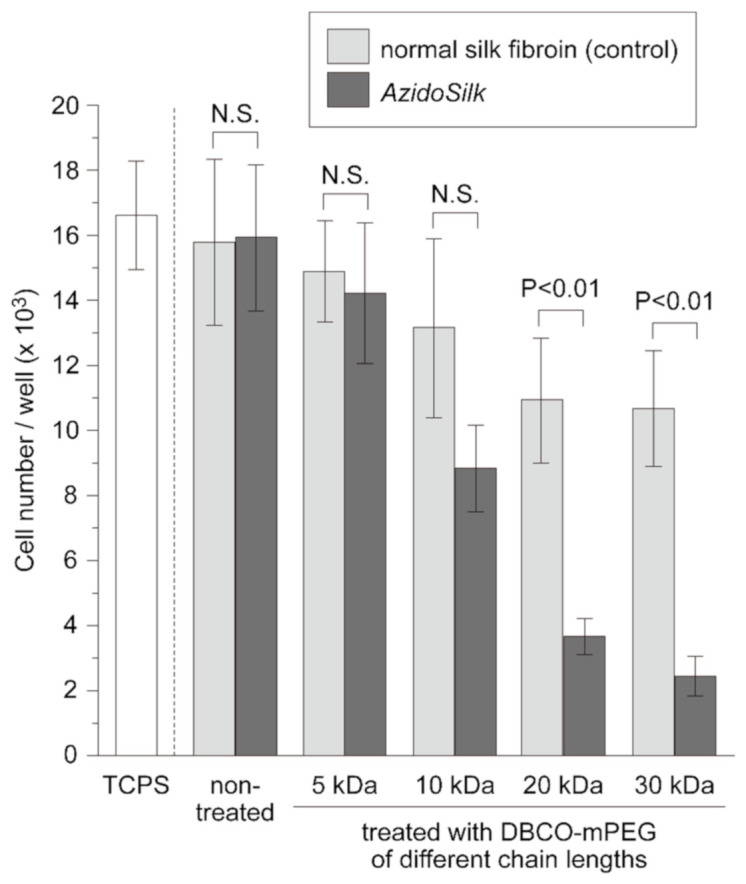
Number of adherent NIH3T3 fibroblasts on the surfaces of normal silk fibroin and *AzidoSilk* films when treated with DBCO-mPEG of different chain lengths. Fibroblasts were cultured for 2 h at 37 °C in Eagle MEM (minimum essential media) without FBS (fetal bovine serum). Cells were counted by the lactate dehydrogenase (LDH) method and the results were averaged. The error bars represent standard deviation (*n* = 8 for TCPS and *n* = 4 for the others). Statistical significance by the Tukey–Kramer test is shown. N.S. denotes no statistical significance (*p* > 0.05).

**Figure 5 molecules-25-04106-f005:**
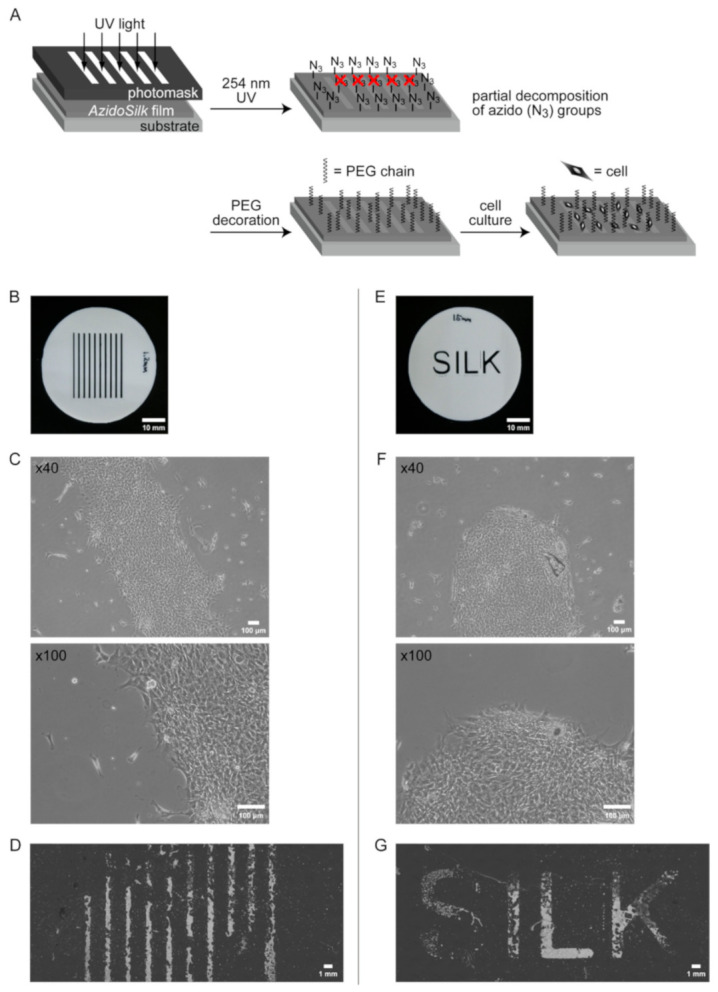
Spatial patterning of NIH3T3 fibroblasts on *AzidoSilk* film modified with DBCO-mPEG of 20 kDa chain length after partial irradiation of 254 nm UV light through patterned photomasks. Fibroblasts were cultured for 3 days in Eagle MEM supplemented with 10% FBS. (**A**) Experimental procedure. UV light was first irradiated to *AzidoSilk* film through a photomask for the partial decomposition of azido groups. After the PEG chain decoration, cells were cultured on the *AzidoSilk* film. (**B**,**E**) Photomasks used in this study. Scale bars are 10 mm. (**C**,**F**) Fibroblasts on the boundary of UV-irradiated and nonirradiated areas on the third day of culturing. Scale bars are 100 μm. (**D**,**G**) Wide views showing patterned adhesion of fibroblasts. Culture medium was removed, and cells were immobilized by 100% methanol followed by drying. Cells turned into white after fixation. Scale bars are 1 mm.

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
