# Peer review of "Click Decoration of Bombyx mori Silk Fibroin for Cell Adhesion Control"

_molecules, 2020, doi:10.3390/molecules25184106_

Round 1
Reviewer 1 Report
The authors report on a functionalization of silk using PEG groups. The PEG-decorated films demonstrated decreased cell adhesion depending on PEG chain length. The idea of using SPAAC for functionalization can be interesting.
One concern is that this paper appears to replicate to a large extent the work reported by same group in a prior work (Int J Mol Sci. 2019 Feb; 20(3): 616) which has however, not been cited in this work. While this is being submitted to a themed issue, the advance of this work is not particularly evident. The authors should clarify what the advance reported herein.
Can the authors (in either this study or prior) comment on the biocompatibility of the AzidoSilk in comparison to a normal silk fibroin?
The control experiment consisted of "normal silk fibroin" treated with DBCO-mPEG of different chain lengths. How was this attached to the silk fibroin over the course of the cell culture? Section 3.6 discusses the procedure, but it is not clear how the attachment was verified.
Was the presence and amount of the PEG assessed prior to and after the cells were cultured? A Live/Dead Assay may also be useful to understand if the surfaces were conducive to the cell culture (currently they do not appear to be proliferating - Figure 3). Compare for instance, morphology shown in Figure 5. Please clarify.
The technique of spatial patterning is not very clear to follow. What kind of mask was used (positive or negative). An indication on the figure where the material is removed/added (or in supporting information) would be helpful. Likewise, spatial patterning of cells on silk fibroin has been previously demonstrated in several works including (Adv Mater 2013, 25 (43), 6207-6212), (ACS Biomater. Sci. Eng. 2020, 6, 1, 705–714), (PNAS 2020 117 (27) 15482-15489). In comparison, Figure 5 does not clearly show the accurate or uniform patterning of cells. These experiments should be clearly shown, including a high resolution panel (c, f) so that the researchers in the field may be able to utilize this technique in comparison to others. Why did the cells turn white following fixation? Figure S3, panels A and B may be better annotated.
This assertion "The same strategy could be extended to other functional molecules such as drugs, polymers, peptides, and enzymes" should be elaborated.
Author Response
<To the reviewer 1>
(1) One concern is that this paper appears to replicate to a large extent the work reported by same group in a prior work (Int J Mol Sci. 2019 Feb; 20(3): 616) which has however, not been cited in this work. While this is being submitted to a themed issue, the advance of this work is not particularly evident. The authors should clarify what the advance reported herein.
Response: Our previous paper (Int J Mol Sci. 2019 Feb; 20(3): 616) reported scaled-up production of AzidoSilk by generating an F1 hybrid of the H06 transgenic line and a high silk-producing strain. The main conclusion of the previous paper was the successful scaled-up production of AzidoSilk by simple hybridization without the decrease of AzPhe incorporation efficiency. On the other hand, in the present manuscript, we are discussing that click decoration of AzidoSilk would be effective to obtain silk fibroin materials with altered cell adhesion properties, although both studies use the same transgenic line, H06. We believe that the importance of the present achievement, cell adhesion control of silk fibroin by click decoration, will be enhanced when combined with the previous achievement because large scale production of cell culture substrates based on AzidoSilk would be possible. To emphasize this point, additional explanation with the reference to the previous paper was inserted in the Introduction (Lines 68-71).
(2) Can the authors (in either this study or prior) comment on the biocompatibility of the AzidoSilk in comparison to a normal silk fibroin?
Response: The biocompatibility of AzidoSilk has not been verified yet. Considering the low content of AzPhe in silk fibroin, approximately two residues per one fibroin molecule comprised of ~5,500 amino acids (mentioned in the 1st paragraph of the section 2.1), we expect that the incorporation of such small number of AzPhe might not cause serious adverse effects on the biocompatibility of silk fibroin. Actually, no significant differences of cell adhesion between on normal silk fibroin and AzidoSilk film (Figure 3) as well as good cell proliferation on AzidoSilk film (Figure S4C) indicate no cell toxicity of AzidoSilk. Additional discussion of this point was inserted (Lines 200-202).
(3) The control experiment consisted of "normal silk fibroin" treated with DBCO-mPEG of different chain lengths. How was this attached to the silk fibroin over the course of the cell culture? Section 3.6 discusses the procedure, but it is not clear how the attachment was verified.
Response: We prepared cell culture dishes coated with normal silk fibroin as controls and subjected them to the same reaction procedure with DBCO-mPEG as with AzidoSilk followed by thorough washing. We assumed that some DBCO-mPEG molecules remained on the surface of normal silk fibroin due to nonspecific binding. However, the binding of DBCO-mPEG was not clearly observed with water contact angle measurements (Figure S2). To clarify this point, we inserted additional explanation on possible nonspecific binding (Lines 158-161).
(4) Was the presence and amount of the PEG assessed prior to and after the cells were cultured? A Live/Dead Assay may also be useful to understand if the surfaces were conducive to the cell culture (currently they do not appear to be proliferating - Figure 3). Compare for instance, morphology shown in Figure 5. Please clarify.
Response: We measured water contact angles to verify the existence of PEG chains on silk fibroin surfaces after click reactions (Figure S2). However, it was hard to determine the amount of PEG on the surfaces quantitatively at this study. We will try to design the experiments for measurements of the PEG amount on the surfaces as the next study. Thank you for your advice. Although, as your advice, Live/Dead assay will be useful to study the properties of the surfaces for cell culture, we aimed to make the surface controllable for cell patterning in this study. The cell adhesion assay in Figures 3 and 4 just observed initial attachment of cells in 2 hours in serum-free media. On the other hand, the cell patterning test in Figure 5 employed longer culturing time for 3 days in the presence of 10% FBS. Such differences in culturing conditions might have resulted in the difference in the cell morphology. We inserted additional explanations on assay conditions (Lines 127-128) and morphology (Lines 129-130).
(5) The technique of spatial patterning is not very clear to follow. What kind of mask was used (positive or negative). An indication on the figure where the material is removed/added (or in supporting information) would be helpful.
Response: The photomask was negative type where azido groups in the exposed areas lost their reactivity with DBCO-mPEG thus retained intrinsic cell adhesion property of silk fibroin. We inserted additional explanations on the photomask (Lines 185-187) and the interpretation of the results (Lines 194-196). A figure explaining experimental procedure was newly added (Figure 5A).
(6) Likewise, spatial patterning of cells on silk fibroin has been previously demonstrated in several works including (Adv Mater 2013, 25 (43), 6207-6212), (ACS Biomater. Sci. Eng. 2020, 6, 1, 705–714), (PNAS 2020 117 (27) 15482-15489). In comparison, Figure 5 does not clearly show the accurate or uniform patterning of cells. These experiments should be clearly shown, including a high resolution panel (c, f) so that the researchers in the field may be able to utilize this technique in comparison to others. Why did the cells turn white following fixation? Figure S3, panels A and B may be better annotated.
Response: Thank you very much for suggesting previous excellent works on cell patterning using silk fibroin. These previous works discuss on the patterning of silk fibroin itself. On the other hand, the present work does not focus on the patterning of fibroin itself but focusing on the patterning of functional molecules directly on fibroin substrate. One advantage of our approach is that we could use self-standing substrates such as film made exclusively of silk fibroin for cell patterning. We inserted an additional paragraph to discuss this point at the end of the Results and Discussion with additional references (Lines 215-225). We fixed and dried cells by methanol which made cells white in order to obtain clear images with low magnification. (Figures 5D and 5G). We are very sorry for the low resolution of these wide view pictures, but we believe that Figures 5C and 5F (and Figure S4B) provide high resolution pictures of cell patterning.
(7) This assertion "The same strategy could be extended to other functional molecules such as drugs, polymers, peptides, and enzymes" should be elaborated.
Response: Click chemistry reactions using azido groups as chemical handles has a wide scope, that is, virtually any kinds of functional molecules are available for click chemistry reactions. Therefore, we expect that other functional molecules such as drugs, polymers, peptides, and enzymes could be spatially patterned on silk fibroin film using the same method reported in the present paper. The above sentence was modified with additional references (Lines 325-327).
Reviewer 2 Report
The authors present a very interesting and timely study of click decoration of B.mori fibroin for cell adhesion control. The reviewer found especially the approach of genetic code expansion interesting and promising.
The short paper is written well, and the entire design is set up quite well.
As a small correction, the authors could make it more clear (in both the text and the figure caption) that for the experiments shown in Figure 2 the DBCO-mPEG concentration was systematically increased (and why).
Author Response
<To the reviewer 2>
(1) As a small correction, the authors could make it more clear (in both the text and the figure caption) that for the experiments shown in Figure 2 the DBCO-mPEG concentration was systematically increased (and why).
Response: We inserted additional explanation in the text (Lines 104-105) and in the figure caption (Lines 119-120).
Reviewer 3 Report
The manuscript entitled “Click Decoration of Bombyx mori Silk Fibroin for Cell Adhesion Control” by Teramoto, Shirakawa, and Tamada describes how an azide-functionalized silk fibroin material produced through bio-engineering the genetic code, can be modified with functional groups to control its cell adhesion properties. This paper continues the story started by the lead author in 2016 (ACS Biomater. Sci. Eng. 2016, 2, 251-258 ) where it was demonstrated that their silk fibroin molecules obtained from transgenic B. mori silkworms containing unnatural azide-bearing phenylalanine amino acids (“AzidoSilk“) could be modified with click chemistry, processed into materials, and photopatterned. In this work, the authors investigate the cell adhesion properties of thin films made from AzidoSilk modified with DBCO-functionalized PEG chains of different lengths by strain-promoted azide-alkyne cycloaddition reactions. The authors point out that the mild conditions and minimal reagents required for this reaction make it ideal for biomedical applications. When treated with PEG chains of length 20 kDa or higher, the AzidoSilk films become significantly less facilitative of mouse fibroblast cell adhesion. By pre-treating AzidoSilk films with UV light through a photomask prior to cell culture, only the non-irradiated areas are amenable to the click reaction; cells grown on these patterned films preferentially attach and proliferate in the UV-irradiated areas.
The experiments are well-designed, the conclusions are justified by the results, and the manuscript is well-written. I recommend this manuscript for publication in Molecules. However, I recommend that the authors make minor revisions addressing the following concerns:
The authors cite the previous work of the lead author (ref 16) to support their claim that the AzidoSilk used in this study was estimated to contain 2 AzPhe residues per molecule. The authors should clarify how they obtained this estimate of the incorporation ratio.
Page 7, line 156: change “Aromatic azido” to “Aromatic azides” or “Aromatic azido groups”
Page 7, line 161, subject-verb and singular-plural agreement: change “AzidoSilk film” to “Azidosilk films” and change “was” to “were”
Author Response
<To the reviewer 3>
(1) The authors cite the previous work of the lead author (ref 16) to support their claim that the AzidoSilk used in this study was estimated to contain 2 AzPhe residues per molecule. The authors should clarify how they obtained this estimate of the incorporation ratio.
Response: We inserted an additional explanation on how we estimated the incorporation ratio of AzPhe (Lines 89-91).
(2) Page 7, line 156: change “Aromatic azido” to “Aromatic azides” or “Aromatic azido groups”
Response: We changed “Aromatic azido” to “Aromatic azido groups” (Line 175).
(3) Page 7, line 161, subject-verb and singular-plural agreement: change “AzidoSilk film” to “Azidosilk films” and change “was” to “were”
Response: We corrected as you pointed out (Line 180).
Reviewer 4 Report
The manuscript by Teramoto et al. reports on production of azido-functionalized silk fibroin and its click chemistry reaction with polyethylene glycol (PEG) to obtain silk fibroin films with tailored cell adhesion properties. The study relies on application of the genetic code expansion methodology in B. mori developed previously in the authors’ laboratory, which allows silk fibroin functionalization via strain-promoted azido-alkyne cycloaddition. The novelty of the present paper is the use of PEG as a substrate for the click reaction and evaluation of the PEG-decorated silk fibroin cell adhesion properties.
In their work, the authors determine the optimal reactant concentrations for the click reaction, study the effect of PEG chain length on cell adhesion properties as well as investigate spatial patterning of cells after partial decomposition of the azido function by UV irradiation using photomasks. The results are important for expanding the biomedical applications of silk fibroin and particularly for the fabrication of various tissue models.
I think that the paper is appropriate for the Molecules special issue "Silk Fibroin Materials" as it reports production and evaluation of novel silk fibroin-based material with tuned cell adhesion properties. The study has been well designed and is clearly described in the manuscript. However, the authors should address the following minor issues before the paper can be accepted:
- Page 1, line 16: please rephrase the sentence “In this study, we applied azido-functionalized silk fibroin, AzidoSilk, produced by a state-of-the-art biotechnology, genetic code expansion, to control the cell adhesion property of silk fibroin film” as it makes the impression that azido-functionalization by itself allows controlling the cell adhesion property of silk fibroin film, which is misleading.
- Several abbreviations found in the manuscript lack definition at their first use, for example, DBCO, mPEG, BmPheRS-a, HRP, CBB.
- Figure 2 caption: please indicate the concentration of the AzidoSilk used in the reactions.
- Page numbers of the Supporting Information are indicated incorrectly.
Author Response
<To the reviewer 4>
(1) Page 1, line 16: please rephrase the sentence “In this study, we applied azido-functionalized silk fibroin, AzidoSilk, produced by a state-of-the-art biotechnology, genetic code expansion, to control the cell adhesion property of silk fibroin film” as it makes the impression that azido-functionalization by itself allows controlling the cell adhesion property of silk fibroin film, which is misleading.
Response: We revised the sentence to “In this study, we applied azido-functionalized silk fibroin, AzidoSilk, produced by a state-of-the-art biotechnology, genetic code expansion, to produce silk fibroin decorated with cell-repellent polyethylene glycol (PEG) chains for controlling the cell adhesion property of silk fibroin film” (Lines 17-18). The sentence to introduce PEG was deleted to avoid redundancy (Line 20-22).
(2) Several abbreviations found in the manuscript lack definition at their first use, for example, DBCO, mPEG, BmPheRS-a, HRP, CBB.
Response: We added definitions of BmPheRS-a (Lines 82-83), DBCO-mPEG (Line 100), HRP (Lines 107-108), MEM (Line 141), FBS (Line 141), LDH (Line 153), and CBB (Supporting Information).
(3) Figure 2 caption: please indicate the concentration of the AzidoSilk used in the reactions.
Response: We inserted additional explanations on the concentration of AzidoSilk and its molar ratios to DBCO-mPEG in the reactions in the Materials and Methods (Lines 271-276, 291-293, 307-308). The molar ratios between AzidoSilk and DBCO-mPEG were sufficiently excessive in all the reactions conducted in the present study.
(4) Page numbers of the Supporting Information are indicated incorrectly.
Response: We corrected the page number in the Supporting Information.
Reviewer 5 Report
The present manuscript by Teramoto et al. demonstrates the novel azido-functionalized silk fibroin films for analysing cell adhesion on surfaces. The study is interesting however, it requires further improvement in terms of characterization in order to meet the high standards of this journal. Please find below my comments to authors:
- Preparation of AzidoSilk as demonstrated by the authors relied on the feeding of the larvae and does not clearly indicate how the AzPhe incorporated in the recombinant peptide sequence. I am aware that the authors have mentioned the previously published protocol (Ref. 16) however, a clear explanation with reference to the previous work should be included for the clear understanding of the readers.
- Partial decomposition of azido groups in the films upon UV illumination needs to be clearly characterized in terms of no. of groups decomposed and remaining.
- Physical immobilization on the surfaces needs to be systematically characterized using FTIR and XPS after each step.
- Since the molecules were surface immobilized physically, I would recommend authors to analyse their stability post modification.
- Effect of surface properties for cell adhesion studies need to be demonstrated and compared with respect to recently published papers. (i) ACS Biomater. Sci. Eng. 2019, 5, 10, 5240–5254 (ii) Langmuir 2018, 34, 11, 3494–3506 (iii) Applied Surface Science, 2020, 505 (1), 144611 (iv) Scientific Reports 4, 4745 (2015).
Author Response
<To the reviewer 5>
(1) Preparation of AzidoSilk as demonstrated by the authors relied on the feeding of the larvae and does not clearly indicate how the AzPhe incorporated in the recombinant peptide sequence. I am aware that the authors have mentioned the previously published protocol (Ref. 16) however, a clear explanation with reference to the previous work should be included for the clear understanding of the readers.
Response: We inserted an additional explanation on how AzPhe is incorporated into protein synthesis in the transgenic silkworms (Lines 83-87).
(2) Partial decomposition of azido groups in the films upon UV illumination needs to be clearly characterized in terms of no. of groups decomposed and remaining.
Response: We irradiated AzidoSilk film with 254 nm UV light for 1.5 min. The irradiation time was decided by an independent UV irradiation experiment using fibroin aqueous solution. The experiment showed that 1 min irradiation was enough to decrease signals from azido groups to background level. The experimental data was added to the Supporting Information (Figure S3). Corresponding additional explanation was inserted to the text (Lines 183-187).
(3) Physical immobilization on the surfaces needs to be systematically characterized using FTIR and XPS after each step.
Response: We did not conduct spectroscopic analyses to verify the immobilization of PEG chains on the surface of AzidoSilk film. Instead, we followed the consumption of azido groups in click chemistry reactions to assess the covalent binding between azido groups in AzidoSilk and DBCO-mPEG (Figure 2). Also, water contact angles were measured to verify the presence of PEG chains on AzidoSilk film (Figure S2). We believe that these data demonstrate the immobilization of PEG chains on AzidoSilk film.
(4) Since the molecules were surface immobilized physically, I would recommend authors to analyse their stability post modification.
Response: We expect that the covalent linkage (triazol bond) formed by click chemistry reactions would be sufficiently stable in physiological conditions. Actually, as we observed clear boundary in Figure S4B after 3 days culture, PEG chains on the film surface remained stable in MEM. However, we recognize that the stability of the materials should be clarified especially when they are going to be applied to in vivo studies.
(5) Effect of surface properties for cell adhesion studies need to be demonstrated and compared with respect to recently published papers. (i) ACS Biomater. Sci. Eng. 2019, 5, 10, 5240–5254 (ii) Langmuir 2018, 34, 11, 3494–3506 (iii) Applied Surface Science, 2020, 505 (1), 144611 (iv) Scientific Reports 4, 4745 (2015).
Response: Thank you for the precious inputs. As related to a comment from another reviewer, we inserted an additional paragraph to discuss the comparison with previous works on the patterning of silk fibroin at the end of the Results and Discussion with additional references (Lines 215-225).
Round 2
Reviewer 1 Report
The paper has addressed the points from the previous round of review and may be accepted.